# Risk Factors for Contracting COVID-19 and Changes in Menstrual and Sleep Cycles in Japanese Female Athletes during the COVID-19 Pandemic

**DOI:** 10.3390/sports10080114

**Published:** 2022-07-28

**Authors:** Yuka Tsukahara, Yuka Hieda, Satomi Takayanagi, Aleksandra Macznik

**Affiliations:** Faculty of Physical Education, Tokyo Women’s College of Physical Education, Tokyo 186-8668, Japan; twcpe181533@outlook.com (Y.H.); s-takayanagi@twcpe.ac.jp (S.T.); alex.evidence.strong@gmail.com (A.M.)

**Keywords:** body mass index, weight gain, vaccination

## Abstract

Although research on COVID-19 is prevalent, risk factors for contracting COVID-19 and lifestyle changes in athletes during the COVID-19 pandemic have not been thoroughly investigated. This study included 254 female collegiate athletes and 107 female non-athletes from Japan, who completed an anonymous survey comprising questions about COVID-19, personal background and lifestyle changes during the pandemic. A total of 6.30% athletes and 6.54% non-athletes had tested positive for COVID-19. The majority reported no change in menstrual cycle (80.31% and 78.50% for athletes and non-athletes, respectively). Wake-up time and bedtime were delayed in some athletes (42.13% and 39.25%, respectively) and non-athletes (46.73% and 31.30%, respectively) during the pandemic. Body mass index (BMI) was significantly higher in COVID-19 (+) athletes than in their COVID-19 (−) counterparts (22.78 ± 3.40 and 21.26 ± 2.06, respectively, *p* < 0.01) and logistic regression analysis revealed that younger students and those with higher BMI had an increased risk of contracting COVID-19. The proportion of vaccinated students was significantly higher in athletes than in non-athletes (*p* < 0.01). Whether the student was an athlete or not was not a related factor in contracting COVID-19. Extra attention should be focused on athletes experiencing weight gain or staying up late and experiencing lower quality sleep.

## 1. Introduction

SARS-CoV-2 (the virus that causes coronavirus disease 2019 (COVID-19) infection) is a highly pathogenic human coronavirus discovered in late 2019, making COVID-19 one of the most dangerous pandemics in the last century. In Japan, patients with COVID-19 were first diagnosed on 16 January 2020. On 30 January 2020, the World Health Organization declared COVID-19 as a pandemic health emergency. Consequently, the Japanese government declared a state of emergency from 7 April to 25 May to restrict people’s movement.

The spread of COVID-19 and its related restrictions have affected athletes in various ways. Despite Japan’s low COVID-19 death rate, many sports events have been cancelled or postponed, including the 2020 Tokyo Olympics and Paralympics, affecting athletes professionally [1]. However, little is known about how to protect athletes from COVID-19. Behaviors such as speaking loudly or shouting, being in physical contact with others, and not wearing masks—all of which occur regularly in sports—are considered risk factors for contracting COVID-19. Several studies reported that intense training can suppress mucosal immune parameters and concluded that playing intense sports could contribute to contracting COVID-19 [2,3]. In contrast, smoking and obesity—considered risk factors for COVID-19—are less likely to occur in athletes than in the general population [4,5,6], and exercise and sports play important roles in improving societal health, particularly when combined with proper nutrition [7]. Thus, to facilitate sports performance in a safe environment, individuals that engage in sports must identify the ideal way to do so, while avoiding the spread of COVID-19.

In addition, because the pandemic has altered people’s lifestyles in various ways, sleep patterns may have been affected. Sleep affects athletes in various ways, including athletic performance and risk for sustaining injury, and although research on changes in the sleep habits of athletes during the COVID-19 pandemic is increasing, further research is necessitated to optimally support athletes [8,9,10]. In South Africa, 79% of athletes reported that their sleep patterns had changed, but this may vary by country because governmental restrictions during the COVID-19 pandemic differed among countries [11,12]. McNamara et al. reported changes in menstrual cycles in athletes during the COVID-19 pandemic, with one in every four elite athletes reporting a change [13]. However, multiple factors affect the menstrual cycle, warranting a systematic enquiry that pertains to different subjects [14,15]. To the best of our knowledge, no study has been conducted on the menstrual cycle in Japanese female athletes participating in multiple sports during the pandemic.

This study aimed to investigate whether participation in sports is a risk factor for COVID-19 infection, as well as to identify the factors that put athletes at risk and those that could protect them from contracting COVID-19. We also aimed to investigate any changes in sleep and menstrual cycles experienced by athletes during the pandemic and identify any differences between athletes and non-athletes. We hypothesized that athletes would contract COVID-19 at a higher rate than non-athletes, owing to the increased contact with other athletes and additional stress associated with changes in their training schedule, resulting in greater changes in their menstrual cycles than those experienced by non-athletes [16,17].

## 2. Materials and Methods

We conducted a cross-sectional study on Japanese female college students that major in physical education, many of whom are athletes, using an online anonymous survey. This study was conducted at the authors’ affiliated institution and abided by the principles of the Declaration of Helsinki. The Ethics Review Procedures Concerning Research with Human Subjects Group of the authors’ affiliated institution approved the study (approval number: 2021-28).

### 2.1. Survey

The survey comprised questions about the participants’ personal characteristics (date of birth, height, body weight, and body mass index (BMI)). Because smoking and certain medical issues (immunocompromise, asthma, etc.) have been reported to be risk factors for worsening COVID-19 symptoms, we also included questions about their medical history, excluding injuries [4,6,18]. Based on a study of prisoners where dormitory residents had higher infection rates than cell residents, we also asked the participants about their housing situations, including whether they lived alone, with family or close friends, or in dormitories [19]. In addition, because surgical masks are reported to prevent transmission, we asked the participants whether they wore surgical masks or non-surgical masks [20]. We further asked them whether they carried hand sanitizers and whether they wore masks during meals and training. For athletes, we asked what type of sport they specialized in and divided them into those playing team or individual, indoor or outdoor, and contact or non-contact sports. We also asked athletes whether they were competing at a national level, how many training days they had per week, and whether they wore masks at all times during training.

We then asked whether they had experienced any change in their menstrual cycle and/or sleep patterns during the pandemic, including changes in sleep quality, sleep duration, wake-up time, and bedtime. Athletes were also asked if they ever came in close contact with someone who had tested positive for COVID-19 and whether they had been vaccinated. The survey was distributed after class in January 2022 and all participants completed it within 10 min of receiving it.

### 2.2. Statistical Analysis

We performed chi-square analysis and Fisher’s exact test to identify differences between athletes and non-athletes, as well as between COVID-19 (+) and COVID-19 (−) groups. To investigate the risk factors for testing positive for COVID-19 and coming in close contact with others, we used multivariate logistic regression to determine the independent variables. We calculated odds ratios (ORs) and associated 95% confidence intervals to determine the strength of the model. Numerical values are listed as means ± standard deviation. All data were analyzed using the Stata 16.1 software (Stata Corporation, College Station, TX, USA). A *p* value of <0.05 was considered statistically significant.

## 3. Results

We received completed surveys from 254 athletes and 107 non-athletes with the response rate of 73.1% (361/494). The survey results are presented in Table 1. There were no significant differences in height, body weight, BMI, and the proportion of students with a history of medical issues between athletes and non-athletes. Regarding housing situation, the proportion of students living in dormitories was significantly higher among athletes than among non-athletes (20.87% vs. 9.35%). Approximately half of the students carried hand sanitizer (53.94% athletes and 46.73% non-athletes) and most used surgical masks (88.93% athletes and 82.86% non-athletes), with no significant difference between the two groups.

A total of 6.30% and 6.54% of athletes and non-athletes, respectively, tested positive for COVID-19, and a total of 18.50% and 21.50% of the athletes and non-athletes, respectively, had come in close contact with a person who tested COVID-19-positive; no significant differences were observed between the two groups. However, the proportion of vaccinated athletes was significantly higher than that of vaccinated non-athletes.

In both groups, the majority reported no changes in menstrual cycle (80.31% and 78.50% for athletes and non-athletes, respectively). Although the majority reported no changes in sleep patterns, sleep quality deteriorated and sleep duration increased in >20% of both athletes (29.13% and 24.80%, respectively) and non-athletes (29.91% and 22.43%, respectively). In addition, some students reported both wake-up time and bedtime delay in athletes (42.13% and 39.25%, respectively) and non-athletes (46.73% and 31.30%, respectively). Among the athletes, 42.13% were competing at a national level, 48.43% participated in outdoor sports, 36.22% participated in contact sports, and 44.16% wore masks at all times during training.

BMI was significantly higher in COVID-19 (+) athletes than in COVID-19 (−) athletes and the proportion of students who had come in close contact with others was higher in COVID-19 (+) athletes (but not significantly) than in COVID-19 (−) athletes. Among non-athletes, none of the factors differed significantly between COVID-19 (+) and COVID-19 (−) non-athletes, except for the change in wake-up time, which did not change for any of the COVID-19 (+) non-athletes. These findings are presented in detail in Table 2.

Results of the logistic regression analysis are presented in Table 3. The main results are as follows:(1)AthletesHigher BMI and younger age were associated with increased risk of contracting COVID-19. Type of sport—outdoor or indoor or group or individual—was not a related factor, and training days per week also had no effect on testing positive for COVID-19.(2)Non-athletesNone of the factors investigated in the survey were relevant in terms of testing positive for COVID-19.(3)All students combinedWith all students considered together, none of the factors investigated in the survey had a relationship with testing positive for COVID-19, and whether the student was an athlete or not was not a related factor (OR 1.03, *p* = 0.96).

## 4. Discussion

In this study, we investigated whether being an athlete is a risk factor for contracting COVID-19 and found that athletes and non-athletes did not differ significantly in terms of contracting COVID-19. This result suggests that, with appropriate protocols, sports can be played safely without increasing the risk of COVID-19 infection. Similar results have been reported in a previous study in hockey athletes [21]. Moreover, despite the fact that playing sports—particularly without wearing masks—can easily lead to close contact, surprisingly in this study, the proportion of students having had close contact with others was lower in athletes than in non-athletes. However, this finding depends on whether athletes would actually be aware or informed of whether an opponent had COVID-19 at the time; thus, further studies are required to address this point. Therefore, our results suggest that playing sports and being an athlete may not necessarily be risk factors for contracting COVID-19 or coming in close contact with other students. Furthermore, although Ahmetov and colleagues reported that playing team sports was a risk factor for contracting COVID-19, our results reflected no difference in the prevalence of students testing positive for COVID-19 between students playing team sports and those playing individual sports [22,23]. However, because certain non-contact team sports may not pose as much of a risk as contact-based team or individual sports, further studies with more subjects participating in both non-contact and contact team sports are required.

It is worth noting that athletes and non-athletes did not differ significantly except in age, which was slightly younger in athletes than in non-athletes, and in vaccination rate, which was higher in athletes. This may be because the coaches had been encouraging the athletes to get vaccinated because it makes it more difficult for athletes to travel if they are not [24,25]. These factors may have influenced the results of this study.

When comparing COVID-19 (+) and COVID-19 (−) athletes, the age was lower and BMI was significantly higher in COVID-19 (+) athletes than in COVID-19 (−) athletes. Although younger age has been associated with a lower risk of developing complications, several studies have identified obesity as a risk factor for patients aged under <60 years [5,26,27,28]. In this study, the BMI of our participants was <25, which is the cutoff value for obesity, but was still associated with a risk of contracting COVID-19. For athletes, a higher BMI could reflect high lean mass rather than obesity; therefore, more research is needed to clarify the effect of BMI on health risks in athletes [29,30]. Turgut and colleagues reported night eating by athletes during the pandemic, which was not investigated in our study [31]. However, it is possible the COVID-19 (+) participants in our study had a similar trend, and if they did eat at night with others, this could have contributed to both contracting COVID-19 and having a higher BMI, as previous studies have reported that eating at night can contribute to weight gain [31,32,33]. In fact, 43.75% of athletes reported that their bedtime had been delayed during the pandemic, compared with the pre-pandemic period. Moreover, age was negatively correlated with contracting COVID-19, indicating the possibility that younger athletes may have had insufficient knowledge on how to protect themselves from COVID-19 infection.

For non-athletes, none of the factors showed significant differences between COVID-19 (+) and COVID-19 (−) students, except for a change in wake-up time because none of the COVID-19 (+) non-athletes reported any changes in their in wake-up time.

Approximately 20% of athletes and non-athletes reported changes in their menstrual cycle during the pandemic, and no significant differences were observed between the two groups. Menstrual cycle changes are commonly caused by stress, and a previous study has reported the relationship between stress and menstrual cycle changes during the pandemic, but the proportion of participants with menstrual cycle changes was lower in our study than in that study [14]. Another study in the US found that 54% of their study subjects reported changes in their menstrual cycles [34]. These figures may have been influenced by cultural differences and varying severity of restrictions between the United States and Japan [34]. It is possible that Japanese athletes experienced lower levels of stress because they had opportunities to go out and exercise, albeit not at pre-pandemic levels.

We found changes in sleep patterns in both athletes and non-athletes, with no significant differences between the groups. In terms of sleep duration, 24.80% and 22.43% of the athletes and non-athletes, respectively, reported that their sleep duration increased compared with pre-pandemic conditions. We speculate that this was because many students were asked to stay home during the pandemic and their wake-up time was consequently delayed, and because sleep quality deteriorated, a longer duration of sleep was required, which was the case for >30% of the athletes and non-athletes.

Our study has several limitations. First, although the overall number of participants was high, the number of students who tested positive for COVID-19, especially among non-athletes, was comparatively low, which could have resulted in statistical weakness.

Second, we did not inquire about the time period over which the sleep duration was proposed to have increased. Third, we did not assess psychological stress; thus, future studies should investigate the relationship between stress and COVID-19, menstrual cycle, and sleep patterns. Finally, because training affects the immune system, owing to stress, future studies should focus on how playing sports affects the immune system and how this may contribute to contracting COVID-19 infection [35,36,37].

Despite the fact that participating in sports involves risk factors for contracting COVID-19, such as physical contact with others, the difference in COVID-19 (+) prevalence between athletes and non-athletes in this study was not statistically significant. A higher BMI and a younger age were associated with athletes testing positive for COVID-19. Therefore, following the pandemic, more attention should be paid to athletes who are experiencing weight gain or staying up late and experiencing poor quality sleep. We believe that because younger students were at higher risk, raising awareness about the risk factors associated with health and healthy behaviors could be beneficial. Changes in menstrual cycles were observed in both athletes and non-athletes, indicating the possibility of psychological stress, a topic worthy of further investigation.

## Figures and Tables

**Table 1 sports-10-00114-t001:** Characteristics, and statistical differences between athletes and non-athletes.

	Athletes (*n* = 254)	Non-Athletes (*n* = 107)	*p* Value
Age (years)	20.62 ± 0.04	20.88 ± 0.13	0.01
Height (cm)	160.15 ± 0.39	159.78 ± 0.53	0.59
Body weight (kg)	54.85 ± 0.44	55.45 ± 0.78	0.48
BMI (kg/m^2^)	21.69 ± 0.27	21.36 ± 0.14	0.22
Medical history, yes (%)	12.60	18.69	0.13
Housing (%)			**0.010**
Live on their own	24.41	20.56	
Live with family or friends	54.72	70.09	
Dormitory	20.87	9.35	
Carry hand sanitizer, yes (%)	53.94	46.73	0.21
Wear mask during meals, yes (%)	75.98	70.75	0.30
Type of masks they use (%)			0.12
Surgical mask	88.93	82.86	
Non-surgical mask	11.07	17.14	
Changes in menstrual cycle (%)			0.35
None	80.31	78.50	
Oligomenorrheic	5.12	9.35	
Dysmenorrheic	4.72	7.48	
Amenorrheic	2.36	0.93	
Polymenorrheic	1.97	0.93	
Multiple symptoms	5.51	2.80	
Sleep quality (%)			
No change	68.50	67.29	0.96
Has deteriorated	29.13	29.91	
Has improved	2.36	2.80	
Sleep duration (%)			
No change	55.51	54.21	0.81
Has shortened	19.29	22.43	
Has lengthened	24.80	22.43	
Undecided	0.39	0.93	
Bedtime (%)			0.72
No change	51.57	47.66	
Has become earlier	6.30	5.61	
Has been delayed	42.13	46.73	
Wake-up time (%)			0.33
No change	54.21	61.42	
Has become earlier	6.54	7.48	
Has been delayed	39.25	31.10	
Smoking, yes (%)	3.94	6.54	0.29
COVID-19 (+), yes (%)	6.30	6.54	0.93
Close contact, yes (%)	18.50	21.50	0.51
Vaccinated, yes (%)	91.34	81.31	**<0.01**
National level (%)	42.13		
Outdoor sports (%)	48.43		
Group sports (%)	45.28		
Contact sports (%)	36.22		
Wear mask at all times during training, yes (%)	44.16		

BMI: body mass index; bold font indicates statistically significant differences between the groups.

**Table 2 sports-10-00114-t002:** Differences between COVID-19 (+) and COVID-19 (−) in athletes and non-athletes.

	Athletes		Difference *p* Value	Non-Athletes		Difference *p* Value
	COVID-19 (+) *n* = 16	COVID-19 (−) *n* = 238		COVID-19 (+) *n* = 7	COVID-19 (−) *n* = 100	
Age	20.51 ± 0.14	20.63 ± 0.05	0.55	20.92 ± 1.34	20.36 ± 0.46	0.28
BMI	22.78 ± 3.40	21.26 ± 2.06	**<0.01**	21.08 ± 2.60	21.73 ± 2.78	0.55
Medical history, yes (%)	12.50	12.61	0.99	42.86	47.00	0.83
Housing (%)			0.68			0.58
Live on their own	25.00	24.37		14.29	21.00	
Live with family or friends	62.50	54.20		85.71	69.00	
Dormitory	12.50	21.43		0.00	10.00	
Carry hand sanitizer, yes (%)	75.00	52.52	0.08	42.86	47.00	0.83
Wear mask during meals, yes (%)	87.50	75.21	0.27	42.86	47.00	0.83
Type of masks they use (%)			0.15			0.84
Surgical mask	100.0	88.19		85.71	82.65	
Non-surgical mask	0.0	11.81		14.29	17.35	
Change in menses (%)			0.87			0.91
None	87.50	79.83		85.71	78.00	
Oligomenorrheic	0.00	5.46		0.00	10.00	
Dysmenorrheic	6.25	4.62		14.29	7.00	
Amenorrheic	0.00	2.52		0.00	1.00	
Polymenorrheic	0.00	2.10		0.00	1.00	
Multiple symptoms	6.25	5.46		0.00	3.00	
Sleep quality (%)			0.36			0.55
No change	56.25	69.33		85.71	66.00	
Has deteriorated	43.75	28.15		14.29	31.00	
Has improved	0.00	2.52		0.00	3.00	
Sleep quantity (%)			0.56			0.34
No change	56.30	43.75		85.71	52.00	
Has become shorter	18.07	37.50		14.29	23.00	
Has become longer	25.21	18.74		0.00	24.00	
Undecided	0.00	0.00		0.00	1.00	
Bedtime (%)			0.56			0.40
No change	56.25	51.26		71.43	46.00	
Has become earlier	0.00	6.72		0.00	6.00	
Has become later	43.75	42.02		28.57	48.00	
Wake-up time (%)			0.33			**0.04**
No change	61.76	56.25		100.00	51.00	
Has become earlier	7.98	0.00		0.00	7.00	
Has become later	30.25	43.75		0.00	42.00	
Smoking, yes (%)	6.25	3.78	0.62			
Close contact, yes (%)	68.75	15.13	**<0.01**			
National level (%)	25.00	43.28	0.15			
Outdoor sports (%)	31.25	49.58	0.16			
Group sports (%)	37.50	45.80	0.52			
Contact sports (%)	50.00	35.29	0.24			
Wear mask at all times (%)	38.46	38.78	0.98			
Training days per week (days/week)	5.19 ± 0.39	5.35 ± 0.09	0.64			
Sanitize their hands during training (times/training)	2.13 ± 0.69	1.85 ± 0.12	0.58			

BMI: body mass index. bold font indicates statistically significant differences between the groups.

**Table 3 sports-10-00114-t003:** Results of logistic regression analysis in athletes, non-athletes, and all students combined.

**COVID-19 (+)** **Athletes**	**Odds Ratio**	**95% CI**	***p* Value**
Age	0.23	(0.06, 0.85)	**0.03**
BMI	1.41	(1.08, 1.84)	**0.01**
Has a medical history	0.26	(0.02, 4.05)	0.34
Smoking	25.73	(0.32, 2066.89)	0.15
Vaccinated	0.69	(0.05, 10.57)	0.79
National level	0.52	(0.10, 2.59)	0.42
Team sports	0.51	(0.10, 2.60)	0.42
Indoor sports	0.41	(0.07, 2.35)	0.32
Contact sports	4.01	(0.85, 18.96)	0.08
Housing			
Live on their own	1.0		
Live with someone	0.29	(0.05, 1.84)	0.19
Dormitory	0.77	(0.08, 6.97)	0.81
Carry hand sanitizer	3.20	(0.62, 16.56)	0.17
Wear mask at all times while eating	2.00	(0.19, 21.06)	0.56
Number of times they sanitize their hands while training	1.23	(0.91, 1.66)	0.17
Training days per week	0.78	(0.47, 1.28)	0.32
**COVID-19 (+)** **Non-Athletes**	**Odds Ratio**	**95% CI**	***p* Value**
Age	0.45	(0.16, 1.29)	0.14
BMI	0.92	(0.69, 1.24)	0.60
Has a medical history	0.63	(0.06, 6.86)	0.71
Smoking	2.30	(0.12, 43.50)	0.58
Vaccinated	0.36	(0.05, 2.83)	0.33
Housing			
Live on their own	1.0		
Live with someone	2.04	(0.17, 23.90)	0.57
Carry hand sanitizer	0.94	(0.14, 6.21)	0.95
Wear mask at all times while eating	0.25	(0.04, 1.44)	0.12
**COVID-19 (+)** **All Students Combined**	**Odds Ratio**	**95% CI**	***p* Value**
Involved in sports	1.03	(0.39, 2.74)	0.96
Age	0.53	(0.28, 1.04)	0.06
BMI	1.15	(0.99, 1.35)	0.07
Has a medical history	0.66	(0.18, 2.42)	0.53
Smoking	3.81	(0.66, 22.11)	0.14
Vaccinated	0.66	(0.16, 2.69)	0.56
Housing			
Live on their own	1.0		
Live with someone	0.97	(0.32, 2.91)	0.95
Dormitory	0.37	(0.06, 2.08)	0.26
Carry hand sanitizer	1.96	(0.75, 5.11)	0.17
Wear mask at all times while eating	0.89	(0.32, 2.47)	0.82

BMI: body mass index. bold font indicates statistically significant differences between the groups.

## Data Availability

The data associated with the paper are not publicly available but are available from the corresponding author on reasonable request.

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
