# Peer review of "Risk Factors for Contracting COVID-19 and Changes in Menstrual and Sleep Cycles in Japanese Female Athletes during the COVID-19 Pandemic"

_sports, 2022, doi:10.3390/sports10080114_

Round 1

Reviewer 1 Report

Thank you for the opportunity to review your article. You have investigated an area of limited research, being the influence of athlete status on COVID risk and the identification of risk factors that may lead to COVID in both athlete and non-athlete populations. The inclusion of considerations for the menstrual cycle should be applauded, as it too represented an under-considered element of female physiology that can often be ignored.

Although the topic and intentions are sound, there are some major points that need to be addressed with this manuscript; all of which are itemised for ease of revision. Overall, the article could benefit from some minor revisions to English language conventions, which are highlighted throughout the document and below. The Introduction contains info that is not cited and presents a hypothesis that is not supported by the evidence presented within the Introduction, despite later being discussed in more detail in the Discussion section. The Methods are inadequate to facilitate replication of the study and does not contain sufficient detail regarding the measures used and how they were collected. The Results presents the findings in large tables for ease of viewing; however many comparisons are missing p-values and results are replicated between the text and the tables. Finally, although the main conclusion surrounding the difference in BMI between athletes who did and did not experience COVID was supported with a significant p-value (p<0.05); this finding and the associated variance between the groups renders the finding to be likely be irrelevant in a practical sense. Furthermore, the discussion of previous literature investigating team sports and the associated increased likelihood of contracting COVID fails to consider contact vs non-contact sports and is not discussed in sufficient detail.

All of these issues can be amended, however are major flaws in the present manuscript in its current form.

Amendments:

Some specific notes for amendment are highlighted through the manuscript and include:

Abstract:

-       There is no mention of Results from the menses component of the investigation in the Abstract. Given the title of the article, please include these findings and conclusion in the Abstract.

-          Please specify that only females were recruited in this analysis, not just “collegiate athletes”.

-          Consider amending the terminology used to describe participants to athletes and non-athletes, as all participants are students.

-          Re-word “The wake-up time and bedtime mostly stayed the same but got delayed in…” to “The wake-up time and bedtime mostly stayed the same or was delayed in…” as it cannot do both stay the same and be delayed.

Introduction:

-          Typo: amend “how protect” to “how to protect”.

-          The statement “Sleep impacts athletes in various ways, including affecting athletic performance and risk for sustaining injury…” requires citation and supporting evidence to substantiate this claim.

-          The hypothesis “…and because of the additional stress of changes in their training schedule, resulting in greater changes in their menstrual cycles than those experienced by non-athletes” is entirely unsupported by evidence presented in the Introduction section. This requires major amendment to ensure hypothesis generated are supported by presented literature.

Methods:

-          Overall, the Methods are inadequate to allow for replication or detailed understanding of the protocols and procedures undertaken during data collection. This is a major limitation of this manuscript. There is insufficient information describing how data was collected and no indication of the timeline, whether this data was collected during the pandemic or following. It is also hard to determine the suitability, reliability or validity of the surveys being used, as no information is provided stating whether a Likert scale used, were the responses on a scale of 1-5 or Less likely to Very likely etc.? These details are key to an appropriate Methods.

Results:

-           Please state what the values and units (for example BMI) in the tables and text represent. For example, are they means and SD or median and confidence intervals?

-          Table 1: there are multiple comparison with missing p-values. These should be included to facilitate a detailed understanding of the findings. Also, for the “Vaccinated, yes (%)” response, the symbol for the p-value appears to be misleading, amend to “p<0.01”.

-          The text appears to replicate p-values and data that is contained within the tables, which is unnecessary, and time should be taken to re-write the text components of the Results.

-          BMI: although there is a significant p-value between COVID (+) and COVID (-) athletes for BMI, it is difficult to see how a 1.52kg/m2 difference is practically important nor impactful, especially as both values are a healthy BMI. This value also does not support the conclusion that athletes undergoing weight gain should be monitored for increased risk of COVID, as there is no analysis or information presented regarding changes in BMI during the pandemic period, only instantaneous cross-sectional data of BMI at a given point in time, which may not have been influenced in any way by the presence of the pandemic.

-          Table 3: the 95%CI for Smoking shows values of [0.32, 2066.89], is this a typo? If so, please amend.

Discussions:

-          The statement “This result suggests that with appropriate protocols, sports can take place safely without increasing the risk of infection with COVID-19“ depends on when this data was collected. If data was collected during the pandemic, many sports and training sessions were conducted under tight protocols or not at all. If taken post-pandemic; yes this would be a suitable statement. This detail is missing from the Methods and needs to be included for clearer interpretation of this conclusion.

-          The statement “… the percentage of students having been in close contact with others was lower in athletes than non-athletes” depends on whether athletes would actually know or be informed whether or not an opponent had COVID at the time of competition. This detail is missing from the Discussion.

-          Team sports and contact sports can be very different in the way players engage and interact. However, contact sports are not clearly discussed as being higher-risk compared to team sports, as there are non-contact team sports that may not results in as larger a risk as contact-based team or individual sports. This should be a discussion point or an additional component of the analysis.

-          The statement “Turgut and colleagues reported night eating by athletes during the pandemic, which was not investigated in our study. However, it is possible that a similar trend was operating in COVID (+) patients in our study, and if they did eat at night with others, this could have contributed to both contracting COVID-19 and having a higher BMI”. How would eating at night contributed to increased risk or factors linked to increased BMI, as there is no inference surrounding energy intake? This point seems largely unfounded and may be stretching the findings to suit conclusion, instead of making conclusion from the results. Also, these are not “patients”, please amend to “participants”.

-          Was any analysis done regarding how much sleep duration was proposed to have lengthened by? If not, this is a minor limitation. Furthermore, this would be easier to interpret with clearer detail regarding the methods, processes and protocols implemented to collect the data.

-          The statement “Second, although it was an anonymous survey, since some students might have felt uncomfortable in answering whether they had tested positive for COVID-19 or not, the number of positive students might therefore have been under-reported. However, this possibility would not affect the differences observed between groups” cold very likely be incorrect. If a participant reported that they didn't have COVID when they really did; the resultant data would be incorrectly classifying that participant to the COVID (-) group instead of the CPOVID (+) group, which could drastically influence the data. Please consider this further and amend the statement surrounding inaccurate self-reporting.

-        The present study findings do not support “Even though playing sports does include risk factors for contracting COVID-19—such as being in physical contact with others” as the p-value presented in Table 3 (p = 0.08) would suggest a non-significant result base don the criteria used in analysis. Please amend this statement to reflect your findings.

Author Response

Thank you for the opportunity to review your article. You have investigated an area of limited research, being the influence of athlete status on COVID risk and the identification of risk factors that may lead to COVID in both athlete and non-athlete populations. The inclusion of considerations for the menstrual cycle should be applauded, as it too represented an under-considered element of female physiology that can often be ignored.

Although the topic and intentions are sound, there are some major points that need to be addressed with this manuscript; all of which are itemised for ease of revision. Overall, the article could benefit from some minor revisions to English language conventions, which are highlighted throughout the document and below. The Introduction contains info that is not cited and presents a hypothesis that is not supported by the evidence presented within the Introduction, despite later being discussed in more detail in the Discussion section. The Methods are inadequate to facilitate replication of the study and does not contain sufficient detail regarding the measures used and how they were collected. The Results presents the findings in large tables for ease of viewing; however many comparisons are missing p-values and results are replicated between the text and the tables. Finally, although the main conclusion surrounding the difference in BMI between athletes who did and did not experience COVID was supported with a significant p-value (p<0.05); this finding and the associated variance between the groups renders the finding to be likely be irrelevant in a practical sense. Furthermore, the discussion of previous literature investigating team sports and the associated increased likelihood of contracting COVID fails to consider contact vs non-contact sports and is not discussed in sufficient detail.

All of these issues can be amended, however are major flaws in the present manuscript in its current form.

Response: We appreciate you taking the time out of your busy schedule for reviewing our manuscript. We would also like to thank you for all of your comments, which have helped us significantly improve the manuscript. We also asked a professional to correct the grammars which we initially did as well but asked for another person to review it.

Amendments:

Some specific notes for amendment are highlighted through the manuscript and include:

Abstract:

-       There is no mention of Results from the menses component of the investigation in the Abstract. Given the title of the article, please include these findings and conclusion in the Abstract.

Response: Thank you for your comment. We have added the following sentence to the Abstract accordingly

ADDED

 The majority reported no change in menses (80.31% and 78.50 % for athletes and non-athletes, respectively).

-          Please specify that only females were recruited in this analysis, not just “collegiate athletes”.

Response: We agree with the Reviewer and have made the following revisions accordingly

FROM

In this study, 254 collegiate athletes and 107 non-athletes from Japan

TO

This study included 254 female collegiate athletes and 107 female non-athletes from Japan

-          Consider amending the terminology used to describe participants to athletes and non-athletes, as all participants are students.

Response: Thank you for your comment. We have amended the terminologies from “athlete students” and “non-athlete students” to “athletes” and “non-athletes,” respectively.

-          Re-word “The wake-up time and bedtime mostly stayed the same but got delayed in…” to “The wake-up time and bedtime mostly stayed the same or was delayed in…” as it cannot do both stay the same and be delayed.

Response: Thank you for your comment, but we meant that both the wake-up time and bedtime remained the same in most athletes but got delayed in other athletes (42.13% and 39.25%, respectively). Thus, we have revised the text as follows:

FROM

The wake-up time and bedtime mostly stayed the same but got delayed in some athlete students (42.13% and 39.25%, respectively)

TO

The wake-up time and bedtime got delayed in some athletes (42.13% and 39.25%, respectively) and non-athletes (46.73% and 31.30%, respectively) during the pandemic.

Introduction:

-          Typo: amend “how protect” to “how to protect”.

Response: Thank you for pointing this out. We have corrected the phrase accordingly as follows:

FROM ‘How protect’ TO ‘How to protect’

-          The statement “Sleep impacts athletes in various ways, including affecting athletic performance and risk for sustaining injury…” requires citation and supporting evidence to substantiate this claim.

Response: Thank you for raising this point. We agree with the Reviewer and have added the following citations in the manuscript: Copenhaver EA et al., 2017, Dwivedi S et al., 2019, and Marshall et al.,2016.

-          The hypothesis “…and because of the additional stress of changes in their training schedule, resulting in greater changes in their menstrual cycles than those experienced by non-athletes” is entirely unsupported by evidence presented in the Introduction section. This requires major amendment to ensure hypothesis generated are supported by presented literature.

Response: Thank you for the comment. We have added the following references to support our hypothesis that additional stress may change the menstrual cycles of athletes: Abplanalp et al., 1977 and Sanders et al., 1999.

ADDED (Abplanalp et al.,1977, Sanders et al., 1999), re-written the paragraph for clarity.

Methods:

-          Overall, the Methods are inadequate to allow for replication or detailed understanding of the protocols and procedures undertaken during data collection. This is a major limitation of this manuscript. There is insufficient information describing how data was collected and no indication of the timeline, whether this data was collected during the pandemic or following. It is also hard to determine the suitability, reliability or validity of the surveys being used, as no information is provided stating whether a Likert scale used, were the responses on a scale of 1-5 or Less likely to Very likely etc.? These details are key to an appropriate Methods.

Response: Thank you for your comments and the survey was distributed after class so the participants have filled them out within 10 minutes. All the possible answers for each question are listed, we did not use Likert scales. More detailed description of the survey was added.

ADDED

The survey was distributed to female students in authors’ institutions in January 2022 during the pandemic. The students filled out a survey after class, which usually took about 5-10 minutes.

Results:

-           Please state what the values and units (for example BMI) in the tables and text represent. For example, are they means and SD or median and confidence intervals?

Response: Thank you for the comment. We used means and SD and have thus revised the text in the Statistical analysis section as follows:

ADDED

Numerical values are listed as means ± standard deviations

-          Table 1: there are multiple comparison with missing p-values. These should be included to facilitate a detailed understanding of the findings. Also, for the “Vaccinated, yes (%)” response, the symbol for the p-value appears to be misleading, amend to “p<0.01”.

Response: Thank you for the comment. Regarding the missing p-values, we performed a chi square analysis on the category, for example, “housing,” and not on all individual responses, such as “live on their own,” “live with family or friends,” and so on. As for the p-value in “Vaccinated, yes (%)” response, that was our mistake and we have corrected it accordingly.rt p-value for “housing”. As for the p-value in “vaccinated” that was my mistake and thank you for pointing that out.

-          The text appears to replicate p-values and data that is contained within the tables, which is unnecessary, and time should be taken to re-write the text components of the Results.

Response: Thank you for the comment. We have removed some of the components that were listed in the tables.

-          BMI: although there is a significant p-value between COVID (+) and COVID (-) athletes for BMI, it is difficult to see how a 1.52kg/m2 difference is practically important nor impactful, especially as both values are a healthy BMI. This value also does not support the conclusion that athletes undergoing weight gain should be monitored for increased risk of COVID, as there is no analysis or information presented regarding changes in BMI during the pandemic period, only instantaneous cross-sectional data of BMI at a given point in time, which may not have been influenced in any way by the presence of the pandemic.

Response: Thank you for the comment. We agree that the BMI values for both COVID (+) and COVID (–) athletes are within the normal range. However, given the proportion of the Japanese population with a normal BMI, a difference of 1.52 kg/m2 is not that small. It is true that this was a cross-sectional study, and we do not know if the COVID-19 pandemic had any effect on BMI values. However, according to previous studies, the normal BMI for collegiate students is below 21.5 kg/m2 and body composition differs between athletes (depending on the sport they play) and non-athletes. Therefore, a BMI of 22.78 kg/m2 is relatively high, which is possibly due to weight gain during the pandemic.

-          Table 3: the 95%CI for Smoking shows values of [0.32, 2066.89], is this a typo? If so, please amend.

Response: Thank you for pointing that out, but given the extremely low number of students who smoke, this is not a typo. We followed the advice of a statistician, and if previous studies have shown that smoking is a risk factor, which it is, it should be included in the model. Fortunately, our study subjects did not include many smokers, which explains the 95%CI values obtained.

Discussions:

-          The statement “This result suggests that with appropriate protocols, sports can take place safely without increasing the risk of infection with COVID-19“ depends on when this data was collected. If data was collected during the pandemic, many sports and training sessions were conducted under tight protocols or not at all. If taken post-pandemic; yes this would be a suitable statement. This detail is missing from the Methods and needs to be included for clearer interpretation of this conclusion.

Response: Thank you for the comment. Because the number of COVID-19 cases remains high in most countries, we believe it is too early to declare a “post-pandemic” period. However, this study was conducted in January 2022, when there were few restrictions in Japan. The study’s main point is that, while there are still many COVID-19 cases, participating in sports may not necessarily increase the number of COVID-19-positive cases if individuals take precautions.

-          The statement “… the percentage of students having been in close contact with others was lower in athletes than non-athletes” depends on whether athletes would actually know or be informed whether or not an opponent had COVID at the time of competition. This detail is missing from the Discussion.

Response: We agree with the Reviewer and we have revised the text as follows:

 “However, this finding depends on whether athletes would actually be aware or informed of whether an opponent had COVID-19 at the time; thus, further studies are required to address this point.”

-          Team sports and contact sports can be very different in the way players engage and interact. However, contact sports are not clearly discussed as being higher-risk compared to team sports, as there are non-contact team sports that may not results in as larger a risk as contact-based team or individual sports. This should be a discussion point or an additional component of the analysis.

Response: We agree with the Reviewer. However, because we included a small number of subjects, further studies with a larger number of subjects are warranted. We have added the following sentence to this effect.

“However, because certain non-contact team sports may not pose as much of a risk as contact-based team or individual sports, further studies with more subjects participating in both non-contact and contact team sports are required.”

-          The statement “Turgut and colleagues reported night eating by athletes during the pandemic, which was not investigated in our study. However, it is possible that a similar trend was operating in COVID (+) patients in our study, and if they did eat at night with others, this could have contributed to both contracting COVID-19 and having a higher BMI”. How would eating at night contributed to increased risk or factors linked to increased BMI, as there is no inference surrounding energy intake? This point seems largely unfounded and may be stretching the findings to suit conclusion, instead of making conclusion from the results. Also, these are not “patients”, please amend to “participants”.

Response: Thank you for the comment. Note that eating at night has been reported as a risk factor for obesity. However, because we had not clarified this point, we have now done so and added a few references. We have also amended “patients” to “participants” accordingly.

ADDED

“Turgut and colleagues reported night eating by athletes during the pandemic, which was not investigated in our study. However, it is possible the COVID (+) participants in our study had a similar trend, and if they did eat at night with others, this could have contributed to both contracting COVID-19 and having a higher BMI, as previous studies have reported that eating at night can contribute to weight gain (Stunkard et al., 1955 et al., Tholin et al., 2009, Turgut et al., 2020).”

-          Was any analysis done regarding how much sleep duration was proposed to have lengthened by? If not, this is a minor limitation. Furthermore, this would be easier to interpret with clearer detail regarding the methods, processes and protocols implemented to collect the data.

Response: We did not ask this question in the survey, so no such analysis was performed. Thank you for pointing that out, and we have added this point to the study limitations as follows:

FROM

“Second, although it was an anonymous survey, since some students might have felt uncomfortable in answering whether they had tested positive for COVID-19 or not, the number of positive students might therefore have been under-reported.”

TO

“Second, we did not inquire about the time period over which the sleep duration was proposed to have increased.”

-          The statement “Second, although it was an anonymous survey, since some students might have felt uncomfortable in answering whether they had tested positive for COVID-19 or not, the number of positive students might therefore have been under-reported. However, this possibility would not affect the differences observed between groups” cold very likely be incorrect. If a participant reported that they didn't have COVID when they really did; the resultant data would be incorrectly classifying that participant to the COVID (-) group instead of the CPOVID (+) group, which could drastically influence the data. Please consider this further and amend the statement surrounding inaccurate self-reporting.

Response: Thank you for raising this point. Because the sentence was confusing, we decided to remove it from the manuscript.

-        The present study findings do not support “Even though playing sports does include risk factors for contracting COVID-19—such as being in physical contact with others” as the p-value presented in Table 3 (p = 0.08) would suggest a non-significant result base don the criteria used in analysis. Please amend this statement to reflect your findings.

Response: Thank you for the comment. We have revised the text as follows:

FROM

“Even though playing sports does include risk factors for contracting COVID-19—such as being in physical contact with others—the difference in COVID-19 (+) prevalence between athletes and non-athletes in this study was not statistically significant”

TO

“Even though playing sports may include risk factors for contracting COVID-19—the difference in COVID-19 (+) prevalence between athletes and non-athletes in this study was not statistically significant which means that with proper precautions, sports could be played without worrying too much of contracting COVID-19.”

Reviewer 2 Report

This study aimed to investigate whether playing sports is a risk factor for becoming infected with COVID-19, but also to identify the factors putting athletes at risk and those that could protect them against contracting COVID-19. Additionally, the authors investigate any changes in sleep and menstrual cycles experienced by athletes amid the pandemic and identify any differences between athletes and non-athletes. The authors hypothesized that athletes could contract COVID-19 at a higher rate than non-athlete-students since they interacted with others more often and because of the additional stress of changes in their training schedule, resulting in greater changes in their menstrual cycles than those experienced by non-athletes.

Considering the last years in which the covid 19 pandemic has dominated the world scene, a study aimed at verifying infections among tleti is certainly of interest. The study appears to be of interest to the scientific and current community. I have only a few minor concerns.

The introduction is well done but perhaps extremely concise. Authors may consider the following work for the introductory paragraph:

- Moscatelli et al., COVI-19: role of nutrition and Supplementation., Nutrients. 2021 Mar 17;13(3):976. doi: 10.3390/nu13030976.

In the paragraph "materials and methods", in the survey section, the authors mentioned injuries. Could you explain this point better? Which types of accidents were taken into consideration and which were excluded and why.

Given that training can affect the immune system and, therefore, facilitate infection or not, in the final paragraph, Discussions, the authors could discuss the effects of training, both short-term and long-term, on the immune system. Additionally, they may also discuss the relationship between training stress and effects on the immune system.  

Author Response

This study aimed to investigate whether playing sports is a risk factor for becoming infected with COVID-19, but also to identify the factors putting athletes at risk and those that could protect them against contracting COVID-19. Additionally, the authors investigate any changes in sleep and menstrual cycles experienced by athletes amid the pandemic and identify any differences between athletes and non-athletes. The authors hypothesized that athletes could contract COVID-19 at a higher rate than non-athlete-students since they interacted with others more often and because of the additional stress of changes in their training schedule, resulting in greater changes in their menstrual cycles than those experienced by non-athletes.

Considering the last years in which the covid 19 pandemic has dominated the world scene, a study aimed at verifying infections among tleti is certainly of interest. The study appears to be of interest to the scientific and current community. I have only a few minor concerns.

Response: We appreciate you taking the time out of your busy schedule for reviewing our manuscript. We would also like to thank you for all of your comments, which have helped us significantly improve the manuscript.

The introduction is well done but perhaps extremely concise. Authors may consider the following work for the introductory paragraph:

- Moscatelli et al., COVI-19: role of nutrition and Supplementation., Nutrients. 2021 Mar 17;13(3):976. doi: 10.3390/nu13030976.

Response: Thank you very much for your advice. We have revised the Introduction section accordingly.

In the paragraph "materials and methods", in the survey section, the authors mentioned injuries. Could you explain this point better? Which types of accidents were taken into consideration and which were excluded and why.

 Response: Thank you for the comment. We decided to ask about asthma, COPD, and diabetes because they could be risk factors for worsening the symptoms of COVID-19. Muscle injuries, on the other hand, would not be a risk factor for COVID-19; thus, muscle injuries were excluded from the survey.

Given that training can affect the immune system and, therefore, facilitate infection or not, in the final paragraph, Discussions, the authors could discuss the effects of training, both short-term and long-term, on the immune system. Additionally, they may also discuss the relationship between training stress and effects on the immune system. 

Response: Thank you for your advice. We have added the following sentence in the limitations:

“Finally, because training affects the immune system owing to stress, future studies should focus on how playing sports affects the immune system and how this may contribute to contracting COVID-19 infection.”